# Identification of Immunoglobulin Gene Rearrangement Biomarkers in Multiple Myeloma through cfDNA-Based Liquid Biopsy Using tchDNA-Seq

**DOI:** 10.3390/cancers15112911

**Published:** 2023-05-25

**Authors:** Natalia Buenache, Andrea Sánchez-delaCruz, Isabel Cuenca, Alicia Giménez, Laura Moreno, Joaquín Martínez-López, Juan Manuel Rosa-Rosa

**Affiliations:** 1Department of Translational Haematology, Research Institute Hospital 12 de Octubre (i+12) Haematological Tumors Clinical Research Unit H12O-CNIO, 28041 Madrid, Spain; natbuena@ucm.es (N.B.); gimenezsa.imas12@correo.h12o.es (A.G.);; 2Department of Translational Haematology, Haematology Service, Hospital 12 de Octubre, 28041 Madrid, Spain; 3Department of Medicine, Faculty of Medicine, Complutense University, 28040 Madrid, Spain; 4Centro de Investigación Biomédica en Red de Cáncer (CIBERONC), 28029 Madrid, Spain; 5Spanish National Cancer Research Center (CNIO), 28034 Madrid, Spain

**Keywords:** liquid biopsy, IG rearrangements, tchDNAseq

## Abstract

**Simple Summary:**

Multiple myeloma has remained largely incurable despite improvements in patient outcomes in the era of targeted anti-myeloma agents. Targeted therapies used in oncology in recent years have significantly changed the way myeloma is treated and thus improved the prognosis for patients. We sought to identify new biomarkers for patient stratification and the prediction of treatment outcomes by applying targeted capture hybridization DNA sequencing (tchDNA-Seq) technology. We have evaluated plasma and bone marrow samples from a homogenous population of 23 patients, which IG rearrangements have the potential to provide important diagnostic, prognostic, and predictive information. We will likely be able to offer a more targeted and risk-adapted therapeutic approach to MM patients at different stages of their disease guided by these potential biomarkers.

**Abstract:**

Multiple myeloma (MM) is a hematological malignancy characterized by the clonal proliferation of pathogenic CD138+ plasma cells (PPCs) in bone marrow (BM). Recent years have seen a significant increase in the treatment options for MM; however, most patients who achieve complete the response ultimately relapse. The earlier detection of tumor-related clonal DNA would thus be very beneficial for patients with MM and would enable timely therapeutic interventions to improve outcomes. Liquid biopsy of “cell-free DNA” (cfDNA) as a minimally invasive approach might be more effective than BM aspiration not only for the diagnosis but also for the detection of early recurrence. Most studies thus far have addressed the comparative quantification of patient-specific biomarkers in cfDNA with PPCs and BM samples, which have shown good correlations. However, there are limitations to this approach, such as the difficulty in obtaining enough circulating free tumor DNA to achieve sufficient sensitivity for the assessment of minimal residual disease. Herein, we summarize current data on methodologies to characterize MM, and we present evidence that targeted capture hybridization DNA sequencing (tchDNA-Seq) can provide robust biomarkers in cfDNA, including immunoglobulin (IG) rearrangements. We also show that detection can be improved by prior purification of the cfDNA. Overall, liquid biopsies of cfDNA to monitor IG rearrangements have the potential to provide important diagnostic, prognostic, and predictive information in patients with MM.

## 1. Introduction

Multiple myeloma (MM) is a highly heterogeneous hematological malignancy characterized by the proliferation of neoplastic plasma cells in bone marrow (BM), which can often progress to extramedullary disease [1]. At the molecular level, MM is characterized by complex cytogenetic and molecular aberrations including chromosomal translocations involving the immunoglobulin (IG) heavy chain locus, copy number variations (CNVs), and somatic mutations in several oncogenic signaling pathways [2,3,4,5,6]. These changes are often associated with increased genomic complexity and have a relevant prognostic impact, which poses a therapeutic challenge [7].

Recent years have witnessed the emergence of novel therapeutic treatments for MM, which have substantially improved disease prognosis [7,8]. Despite the substantially prolonged survival, however, MM remains incurable in most patients and relapses are common. Hence, strategies that allow more assiduous clinical assessment over time are needed to identify early signs of therapeutic resistance and to aid clinicians in selecting the best therapeutic option before relapse occurs [9,10].

The current gold standard for the genetic diagnosis of MM remains cytogenetic profiling of plasma cells from BM samples [11], which provides data on chromosomal abnormalities but not other anomalies. More recently, the monitoring of minimal residual disease (MRD) by flow cytometry of BM samples has become a leading method to evaluate the depth of response to treatment in patients with MM by detecting persisting tumor cells [12]. We believe that a key challenge is to assess whether additional information obtained by molecular karyotyping is helpful in the management of MM. In this regard, we recently developed a new targeted capture hybridization DNA sequencing (tchDNA-Seq) gene panel [13], which combines the identification of different biomarkers in one single method, which can lead to superior genetic characterization of patients.

A liquid biopsy is defined as the detection of circulating tumor cells (CTCs) and/or small fragments of cell-free DNA released from primary and secondary neoplastic lesions [14,15,16,17] into peripheral blood (PB). Therefore, it is considered to offer a minimally invasive method to detect tumor markers and capture tumor heterogeneity [18]. Hybrid capture and targeted sequencing of cfDNA has been reported in hematological cancer genomes [19]; however, whether liquid biopsy can be used to assess genomic alterations in MM in a similar way to baseline tests is still under study. Many studies have shown that cfDNA analysis in MM also appears to accurately detect mutations identified by genetic profiling of bone marrow-derived tumor DNA, with good concordance between paired plasma samples [20,21,22]. We observed that the purification of cfDNA samples from genomic DNA using a specific magnetic beads ratio [23] clearly improves the signal detection performance of monitored biomarkers. We believe this method can potentially replace medically unnecessary BM sampling and provide an alternative non-invasive test for longitudinal genetic monitoring of patients with MM receiving targeted therapy [19].

## 2. Technological Methods in Liquid Biopsy

### 2.1. Present MRD Approaches

Currently, the assessment of MRD in MM is routinely performed by multiparametric flow cytometry (MFC) of BM samples [24,25,26,27]. MFC is a robust technique for the detection of BM plasma cells and can clearly discriminate between normal and clonal plasma cells, even when they are present at very low frequencies, as occurs in MRD monitoring [28,29]. For example, the sensitivity of 8-color MFC is 1 clonal plasma cell in 10^5^ normal cells, although highly standardized protocols from the EuroFlow consortium can reach a sensitivity of 1 in 10^6^ [12]. It has recently been reported that a sensitivity of 10^−7^ can be achieved with a good correlation with BM MRD assessments using a CD138+ cell purification strategy in PB [30].

Sampling of the BM by aspiration is, nevertheless, an invasive and painful procedure for patients, and the use of liquid biopsy, which is much less invasive, has shown great potential in hematological malignancies in recent years [31]. Similarly, next-generation sequencing of IG loci has been developed in the last 5 years to overcome these limitations of conventional MFC techniques for MRD monitoring [12,32,33]. Indeed, NGS increasingly appears to offer an important advance in genetic mapping to detect new aberrations involved in disease progression. The clonoSEQ Assay manufactured by Adaptive Biotechnologies (Seattle, WA, USA) is the first and only next-generation sequencing-based MRD test authorized by the U.S. Food and Drug Administration (FDA) for MRD assessment in patients with acute lymphoblastic leukemia (ALL) or multiple myeloma. This identifies and quantifies gene sequences in DNA extracted from the bone marrow using multiplex polymerase chain reaction and NGS. The assay detects as few as 1 tumor cell within more than 1 million healthy cells, and several studies using this assay for the assessment of patients with MM demonstrated that the MRD level correlates with the outcome and that the lower the MRD level, the better the prognosis [34,35].

### 2.2. Disease Advancement and Relapse

The levels of V(D)J rearrangements in circulating free tumor DNA (ctDNA) correlate with clinical disease activity. Recently, Oberle et al. used NGS to detect the V(D)J clonotypic rearrangement and for subsequent follow-up by liquid biopsy after treatment initiation in MM patients. They observed that the majority of non-responders had traces of persistent myeloma cells. Positivity for CTCs and cfDNA were associated with each other in most cases, but were discordant in 30% of cases, indicating that cfDNA may not be generated entirely by CTCs and may reflect the bulk tumor burden [36]. Similarly, Biancon et al. identified clonal rearrangements of the IGH gene in PPCs and cfDNA samples from MM patients receiving second-line treatment. The same clonal IGH rearrangement identified in PPCs was detected in paired plasma samples, and levels of IGH cfDNA correlated with the outcome [37].

The detection of CNVs is also a clinically relevant approach to characterizing genomic abnormalities in MM. Guo et al. questioned whether CNVs and clonal somatic mutations could be robustly detected across the entire genome or exome using cfDNA in patients with active disease and whether the mutations found in the BM could be reliably reproduced in the cfDNA. Their study demonstrated that the comprehensive genomic interrogation by whole exome and genome sequencing of MM-derived cfDNA is feasible and allows detailed genomic insight into MM evolution and temporal progression [21].

Mutational characterization of MM is currently based on BM biopsy, which does not capture the spatial and genetic heterogeneity of this multifocal disease. Mithraprabhu et al. investigated the clinical utility of using plasma-derived ctDNA for mutational characterization and for tracking disease progression. Paired BM cell DNA and ctDNA from 33 relapsed/refractory and 15 newly diagnosed patients with MM were analyzed for *KRAS*, *NRAS*, *BRAF,* and *TP53* mutations by NGS, which revealed a higher frequency of ctDNA-only mutations in relapsed/refractory disease, authenticating the spatial heterogeneity of MM. Accordingly, ctDNA analysis can be used as an adjunct to BM aspirate, representing a noninvasive strategy for improved mutational characterization and therapeutic monitoring of MM [38]. Along the same line, Rustad et al. explored the relationship between ctDNA and disease activity during long-term follow-up by monitoring recurrent mutations (*NRAS*, *KRAS*, and *BRAF*) with digital droplet PCR and comparing the results with those from the analysis of BM. They observed a good correlation between the concentration of mutated alleles in BM cells and in ctDNA, which reflects mutated cells, the total tumor mass, and the transformation to a more aggressive disease [39]. 

### 2.3. Methodological Approaches

In addition to the type of sample analyzed, it is also important to consider the technique adopted to obtain the greatest possible sensitivity and the most reliable data for patient monitoring. In this respect, a targeted NGS approach might be the most appropriate option. Gerber et al. analyzed the MM mutational landscape in the cfDNA of patients by tracking the clonotypic V(D)J rearrangement as a patient-specific marker, and by genotyping a specific set of genes. For the latter, they designed a targeted panel, including coding exons and splice sites of 14 genes (*BRAF*, *CCND1*, *CYLD*, *DIS3*, *EGR1*, *FAM46C*, *IRF4*, *KRAS*, *NRAS*, *PRDM1*, *SP140*, *TP53*, *TRAF3*, and *ZNF462*), and developed a cancer personalized profiling by deep sequencing (CAPP-seq) approach to compare the mutational profile of cfDNA and tumor genomic DNA of purified plasma cells from BM aspirates of different disease stages, including early disease. They found that the amount of cfDNA correlated with some parameters that may indicate the tumor burden, such as the percentage of plasma cell infiltration of BM [40]. Likewise, Kis et al. applied a hybrid capture-based liquid biopsy sequencing (LB-Seq) method to sequence *KRAS*, *NRAS*, *BRAF*, *EGFR*, and *PIK3CA* in cfDNA specimens from patients with myeloma, concluding that cfDNA analysis is comparable to myeloma molecular profiling using BM tumor cells. Their results revealed a high concordance between both fractions with high specificity, proposing LB-Seq as an implementation for genetic profiling of BM in MM. These findings have been confirmed by other studies [37,38].

Regarding the study of MRD using cfDNA, several reports have compared it with conventional techniques of MRD evaluation in BM, with conflicting results. For instance, Mazzotti et al. reported no correlation between ctDNA and BM for MRD analysis by NGS using only IGH gene rearrangements in patients with myeloma [41]. In a similar context, Long et al. analyzed plasma samples from 8 patients with extramedullary MM and other plasma samples from patients with MM but without extramedullary plasmacytomas. They found that patients with extramedullary disease had higher cfDNA concentrations. They designed sequencing panels targeting the coding sequence regions of 22 recurrently mutated genes, detecting 17 different somatic mutations. They concluded that cfDNA can be used as a surrogate material to track extramedullary disease progression, particularly when plasmacytomas are inaccessible [42]. However, in another study, although a relevant concordance in clonal somatic mutations (~99%) and copy number alterations (~81%) was high, a clinical correlation was not found, and the results in the case of MRD monitoring were not significant [19]. Currently, the use of cfDNA alone has no utility in the assessment of MRD in patients with MM, although, as we state, there is increasing evidence that it could be a useful complementary tool. Indeed, with further refinement cfDNA monitoring could be a relevant modality for the assessment of MRD [42,43]. Deshpande et al. also assessed whether cfDNA levels vary according to risk status using a targeted NGS approach. They found that the cfDNA levels in 77 patients were significantly higher in the high-risk MM group and correlated weakly with clinical markers. Patients with high cfDNA levels were associated with a worse progression-free and overall survival [44]. At this time, however, targeted sequencing of cfDNA cannot achieve the sensitivity of MRD detection by flow cytometry or molecular analysis in BM. Nevertheless, in the future, MRD analysis using cfDNA might have sufficient sensitivity through the use of reliable biomarkers. In this context, we have developed a new targeted method (tchDNA-Seq) combining the following molecular parameters: patient-specific mutation panels, translocations patterns, copy number alterations, and IGH rearrangements. We believe that this approach might increase the accuracy of progression-free survival prediction and the detection of false-negative MRD.

### 2.4. A New Liquid Biopsy Approach: Targeted Capture Hybridization Panel (tchDNA-Seq)

#### 2.4.1. Samples

We analyzed 51 samples obtained at diagnosis from 23 patients: 23 samples of genomic DNA from PPCs, 16 samples of genomic DNA from total BM, and 12 cfDNA samples obtained from PB (the clinicopathological features are described in Table 1 from Rosa-Rosa et al., 2022) [13]. Plasma samples should be stored in tubes that do not affect cfDNA stability or promote cell lysis, which is a major concern. In this respect, we have had good results with the Cell-free DNA blood collection tubes (cfDNA BCT tubes) commercialized by Streck (La Vista, NE, USA). The plasma samples were processed using the QIAamp Circulating Nucleic Acid Kit (Qiagen, Hilden, Germany, cat. no. 1017647). Further, a liquid biopsy for cfDNA was performed across a total of 11 follow-up samples from three patients. One cfDNA sample was purified using AMPure XP (Beckman Coulter Life Sciences, Indianapolis, IN, USA) magnetic bead-based size selection to increase the signal obtained. We removed genomic DNA using a 0.6× concentration of beads, followed by a 1.8× concentration of beads to capture the remaining cfDNA. Isolated cfDNA was analyzed with an electropherogram Agilent 2100 Bioanalyzer (Agilent Technologies, Santa Clara, CA, USA) to check the optimum profile.

#### 2.4.2. Panel Description and Sequencing

We designed a specific panel to investigate the genomic alterations present in patients with myeloma [13]. The panel contains genes involved in the development and progression of MM (*ATM*, *ATR*, *CCND1*, *KRAS*, *NRAS*, *PRDM1*, etc.), genes related to treatment or drug resistance (*PSMD1*, *XBP1*, *PSMB5*, *PSMC2*, etc.) and candidates for new immunotherapy treatments (*CD38*, *BCMA*, *GPRC5D*). A list of all genes is provided in Appendix A. In addition, we also considered genomic regions frequently involved in MM, including the translocations t (4;11), t (4;14), t (4;16), and t (11;14), and regions distributed in the genome for the identification of CNVs and chromosomal regions of the IGH locus. Libraries were generated with SureSelect reagent kits (Agilent Technologies, Palo Alto, CA, USA), using 50 ng of genomic DNA for PPC and BM and 10–200 ng of cfDNA. Sequencing was performed on Illumina NextSeq 500 and/or Ion Torrent platforms. A specific bioinformatics process was applied, which included alignment with the hg38 genome, identification of SNVs and Indels by combining variant calling, and IGH/K rearrangements with MiXCR 3.0.13, a software platform for analysis of NGS data for immune profiling.

## 3. Preliminary Results Using the New tchDNA-Seq

### 3.1. Assay of Genomic DNA and cfDNA Samples at Diagnosis

Analysis of the alterations present in PPCs, genomic DNA, and cfDNA revealed a total of 97 alterations in the 23 PPC samples and at least 1 alteration was present in all samples. Sixty-seven of the alterations found in PPCs were also found in BM DNA and thirty-two in cfDNA. Positively, 1 of the 2 translocations seen in PPCs was traced in cfDNA. The results reveal a concordance between the BM and cfDNA data, detecting samples with at least one alteration in 100% of the cases (Appendix A). Although we can assume lower VAFs correlated with subclonal mutations, no studies of subclonality have been performed globally in this dataset. After an analysis of the results, we concluded that the detection of SNVs in liquid biopsy is limited by signal dilution; specifically, a reduction of approximately 1 log in cfDNA and BM compared with PPCs, whereas the median reduction in frequency observed in cfDNA and BM for IG rearrangements was not significant compared with that observed in PPCs. Moreover, based on the comparison of frequencies of SNVs observed in either cfDNA or BM with that observed in PPCs, we estimated a median tumor burden of 30% and 4% in BM and cfDNA, respectively (Table 1). It is important to mention that in the five cases where PPCs, BM, and cfDNA were available, 55% of the mutations were present in all fractions and similarly, two of the five cases with all three fractions showed the same rearrangement in PPCs, BM, and cfDNA. (Figure 1). No translocations were seen in these patients. In summary, these results confirmed the detection of IG in cfDNA as a robust biomarker of MM.

### 3.2. Liquid Biopsy in Follow-Up Samples

We analyzed tumor cfDNA in patients by tracking the V(D)J clonotypic rearrangement as a fingerprint of the disease and genotyping a set of cancer genes or patient-specific translocations identified at diagnosis. As described above, we performed liquid biopsy at several follow-up time points from the three patients and the results are described below as individual cases (Figure 2).

#### 3.2.1. Patient 1

Presented with rearrangements in the IGH locus and one-point mutation at diagnosis (*IGF1R* p.T677A). No mutations were detected in the two follow-up studies with positive MRD by MFC, but the same IG rearrangement was found in the BM fraction but not in cfDNA (Figure 2A).

#### 3.2.2. Patient 2

At diagnosis, two point mutations (*KRAS* G12D and *DIS3* D479E), t (11,14), and different IG rearrangements were identified. From the four follow-up studies with positive MRD by flow cytometry, IG rearrangements in cfDNA could only be observed at follow-up 2 and follow-up 4. No translocations or point mutations were detected. As a cfDNA signal was detected in this assay, in contrast to patient 1, we wondered whether negative results in cfDNA could be due to purity conditions. A purification test was run on the negative cfDNA sample from follow-up 3 to determine if the signal was restored. The remaining genomic DNA was removed from the cfDNA samples by a specific ratio of magnetic beads according to Chauhan et al. [23]. Subsequently, the IG rearrangement was detected in follow-up 3. Nonetheless, after purification mutations and translocations did not manifest (Figure 2B).

#### 3.2.3. Patient 3

Liquid biopsy was also performed on follow-up samples and the results were compared with those obtained at diagnosis. Interestingly, we observed that in the cfDNA and BM follow-up samples, the signal of the initial mutations disappeared and a *TP53* mutation emerged that had not been considered before. Monitoring of the patient showed that, in the last follow-up and also coinciding with the cessation of treatment, a decrease in the *TP53* allele burden was found. Although the patient is currently in remission, the *TP53* mutation may condition the development of another undetected tumor different from the primary one, so follow-up remains important (Figure 2C).

## 4. Conclusions

Recent advances in genomics and bioinformatics have facilitated the development of precision medicine programs and allowed increasing numbers of solid tumors to undergo rapid whole-genome sequencing (WGS) according to Subhash et al. Whilst the primary focus of precision medicine has been the rapid identification of targetable genetic alterations, the WGS data can also be used to determine unique tumor-specific gene sequences and to develop patient-specific MRD assays, independent of tumor type [45], that could be extrapolated to myeloma patients. These findings highlight the potential to serve as a biomarker to track treatment response and to improve the detection of MRD following disease control.

Most of the molecular alterations manifesting in PPCs appear to be present in the cfDNA of patients, including translocations. The correlation observed between BM samples and PPCs might indicate that no separation of the positive fraction is necessary for sample processing, especially in patients with an infiltration higher than 10%. In addition, regardless of the infiltration of MM in the BM, all molecular alterations have been found at diagnosis. However, the detection of SNVs by liquid biopsy is limited by signal dilution, with an approximately 1 log reduction in cfDNA compared with PPCs, which must be considered in bioinformatics analyses. 

Regarding IGH/K rearrangements, a minor signal reduction was observed, confirming that IG rearrangements are strong biomarkers of MM in cfDNA. Importantly, the purification of cfDNA and removal of the remaining genomic DNA may play an important role in improving results in terms of the ability to detect biomarker signals. We believe that the signal in cfDNA may be diluted and, therefore, purification of cfDNA in liquid biopsy could be important. In addition, we show that new molecular lesions can be identified through a capture panel.

## 5. Future Directions

With regard to future directions, it would be desirable to increase the cohort size to validate these promising results. Since neither point mutations nor translocations could be observed by the panel, our efforts are now directed toward improving the development of tests that will enhance detection. In this line, we have designed new primers to verify if we specifically observe the signal for translocation. Our preliminary results demonstrate that the signal from the translocation is not affected by the bias that primers can produce in a PCR although we are still working on the problems arising from customized primers. Finally, it is a priority to improve the collection of samples in appropriate tubes and conditions, as well as to implement purification methods that have proven to be effective. Although liquid biopsy is promising in MM, it is necessary to improve the sensitivity and accuracy and increase the number of studies.

## Figures and Tables

**Figure 1 cancers-15-02911-f001:**
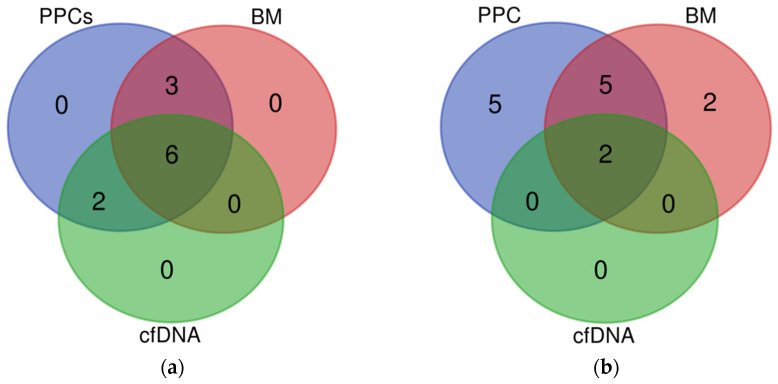
Venn diagrams from capture hybridization sequencing panel of 5 matched PPCs, cfDNA, and BM tumor samples: (**a**) clonal mutations present in PPCs (blue) and identified in either cfDNA (green) or BM biopsies samples (red); (**b**) rearrangements present in PPCs samples (blue) and observed in cfDNA (green) and BM biopsies (red).

**Figure 2 cancers-15-02911-f002:**
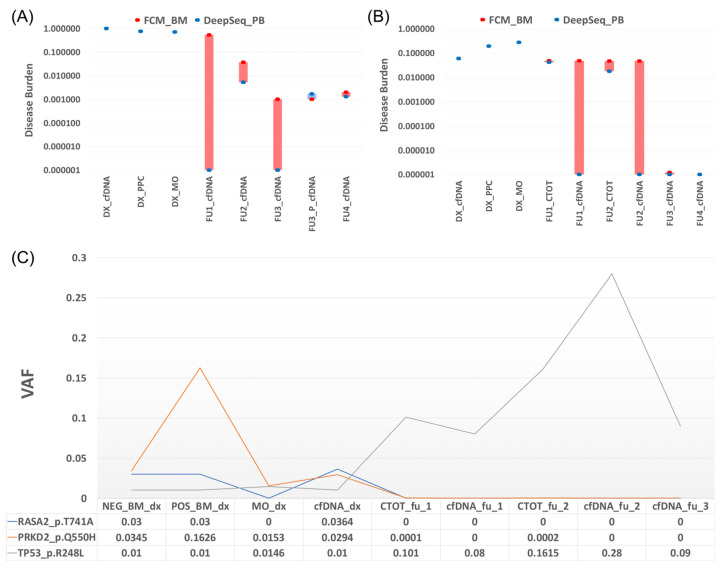
Description of the alterations related to the three analyzed patients by targeted capture hybridization DNA panel. (**A**) Patient 1: comparison of disease burden obtained by FCM (red point) and deep sequencing (blue point). Four positive MRD follow-ups are observed by FCM, and the only alteration is found in the total bone marrow cells (CTOT) fraction of FU2 by tchDNA-Seq. (**B**) Patient 2: comparison of disease burden obtained by FCM (red point) and deep sequencing (blue point). Four positive MRD follow-ups are observed by FCM, and three alterations (FU2, FU3 purified, and FU4) were found in cfDNA. Bars (red and blue) show the differences between the two methods. (**C**) Patient 3: Mutational evolution of different types of samples at diagnosis and follow-up points.

**Table 1 cancers-15-02911-t001:** Comparative results among PPC, BM, and cfDNA. The VAF ratio is obtained by dividing the VAF observed in BM/cfDNA and the VAF observed in PPC for common SNVs and IG rearrangements.

(A) PPC vs. BM Comparison				(B) PPC vs. cfDNA Comparison			
Alteration Type	In PPC	In BM	Median VAF_Ratio	Alteration Type	In PPC	In cfDNA	Median VAF_Ratio
SNVs + IGs	70	51	0.58	SNVs + IGs	43	24	0.06
SNVs	36	32	0.31	SNVs	23	20	0.04
IGs	34	19	0.80	IGs	20	4	1.13 *

Abbreviations: PPC—pathogenic plasma cells; BM—bone marrow; cfDNA—cell-free DNA; VAF—variant allele frequency; SNV—single nucleotide variants; IG—immunoglobulins. * VAF in cfDNA is higher than that observed in PPC at diagnosis.

## Data Availability

The data not presented in this article can be shared up on request.

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
