# Peer review of "Identification of Immunoglobulin Gene Rearrangement Biomarkers in Multiple Myeloma through cfDNA-Based Liquid Biopsy Using tchDNA-Seq"

_cancers, 2023, doi:10.3390/cancers15112911_

Round 1

Reviewer 1 Report

Targeted therapies used in oncology in the last 10-15 years have significantly changed the way cancer is treated and thus changed the life prognosis of patients for the better. Almost all cancers, including blood cancers, acquire resistance to systemic chemotherapy. This is related to the diversity of tumor cells, clonal evolution and the selection of resistant clones caused by anti-cancer treatment.

Although genotyping is currently the most used method of classifying cancer risk for treatment decisions, many researchers are trying to overcome the problems associated with this type of tedious and cost-intensive diagnosis and are trying to identify other methods that are quick, non-invasive, and more cost-effective. The sought method is to meet the identification of biomarkers at different time points in the course of the disease. Since cell-free circulating tumor DNA (ctDNA) is a potential substitute for the entire tumor genome, the use of ctDNA analysis as a liquid biopsy holds promise for helping to obtain urgently needed genetic data.

The draft itself is interesting, and the methodology of work is presented in a clear way. The work is written in a good scientific language based on current literature. 18 out of 44 references come from the last 5 years, of which 4 from the last two years. I believe that the publication of the above data provides interesting information, shows a new stage of cancer diagnosis in a simple way and seems promising for implementation in the clinic. It is also interesting to be able to use it in the monitoring of residual disease, which will probably make it possible to improve the quality of care in oncology in the future.

Reviewer 2 Report

Re: cancers-2382232

Buenache et al summarized current data on methodologies to characterize MM, and they presented evidence that targeted capture-hybridization DNA sequencing (tchDNA-Seq) can provide robust biomarkers in cfDNA, including immunoglobulin (IG) rearrangements. They insisted that detection can be improved by prior purification of the cfDNA. 

I could not get enough information regarding their methods to analyze “Panel descrition and sequencing” and methods of NGS and FCM to detect MRD. What kinds of methods did you use? How about the sensitivity and specificity of MRD detection.

Regarding the “Preliminary Results using the new tchDNA-Seq”, how about the patients’ characteristics? The number of patients seem to be small, but I cannot get this information.

Figure 2: I do not understand the difference between points and bars. What is “CTOT”?

What is “BCT tubes”?

Reviewer 3 Report

I have the following critiques/questions about this manuscript:

1. Figure2 is very poorly presented. The axes labels are very hard to read and interpret. I suggest that the label size be increased and made readable for any meaningful interpretation. 

2. Table1B, IG VAFs are > 1, please include a possible explanation for this if it is correct. 

3. A lot of research is being done to use whole exome/genome sequencing for MRD in solid tumor context. Please add a note about where these technologies fit in the MM context in the Discussion section. 

4. Figure2C the table headings are hard to read and please improve the resolutions

5. Please add a note on whether sub-clonal mutations were detected. If so how were they treated. 

Round 2

Reviewer 2 Report

I have no additional comments.